# Doing What We Know, Knowing What to Do: Californians Linking Action with Science for Prevention of Breast Cancer (CLASP-BC)

**DOI:** 10.3390/ijerph17145050

**Published:** 2020-07-14

**Authors:** Jon F. Kerner, Marion H. E. Kavanaugh-Lynch, Lourdes Baezconde-Garbanati, Christopher Politis, Aviva Prager, Ross C. Brownson

**Affiliations:** 1California Breast Cancer Research Program, Bethesda, MD 20186, USA; 2California Breast Cancer Research Program University of California, Office of the President, Oakland, CA 94612, USA; marion.kavanaugh-lynch@ucop.edu; 3Preventive Medicine, Community Initiatives, Keck School of Medicine (KSOM), University of California, Los Angeles, CA 90033, USA; baezcond@usc.edu; 4Community Engagement, Norris Comprehensive Cancer Center, University of California, Los Angeles, CA 90033, USA; 5Center for Health Equity in the Americas, KSOM, University of Southern California, Los Angeles, CA 90007, USA; 6Cancer Screening, Canadian Partnership Against Cancer, Toronto, ON M5H 1J8, Canada; Christopher.Politis@partnershipagainstcancer.ca; 7California Pan-Ethnic Health Network, Oakland, CA 94612, USA; aprager@cpehn.org; 8Brown School, Washington University in St. Louis, St. Louis, MO 63130, USA; rbrownson@wustl.edu; 9Department of Surgery (Division of Public Health Sciences) and Alvin J. Siteman Cancer Center, School of Medicine, Washington University, St. Louis, MO 63110, USA

**Keywords:** implementation and dissemination, primary prevention, community-based participatory research, breast cancer, population science, action research

## Abstract

Given the lack of progress in breast cancer prevention, the California Breast Cancer Research Program (CBCRP) plans to apply current scientific knowledge about breast cancer to primary prevention at the population level. This paper describes the first phase of Californians Linking Action with Science for Prevention of Breast Cancer (CLASP-BC). The foci of Phase 1 are building coalitions and coalition capacity building through community engagement in community-based participatory research (CBPR) and dissemination and implementation (D&I) research training. Based on the successful implementation and evaluation of Phase 1, the foci of Phase 2 (presented separately in this special issue) will be to translate the California Breast Cancer Prevention Plan overarching goal and specific intervention goals for 23 breast cancer risk and protective factors strategies into evidence-informed interventions (EIIs) that are disseminated and implemented across California. CLASP-BC is designed to identify, disseminate and implement high-impact, population-based prevention approaches by funding large scale EIIs, through multi-jurisdictional actions, with the intent to decrease the risk of breast cancer and other chronic diseases (sharing common risk factors), particularly among racial/ethnic minorities and medically underserved populations in California.

## 1. Introduction 

Breast cancer is the most common cancer in women and the largest cause of cancer deaths among women worldwide: there were an estimated 2.1 million new cases and 626,679 deaths in 2018 [1]. A woman in the USA has a 13% chance of being diagnosed with breast cancer at some point in her lifetime and a 2.6% chance of dying from breast cancer [2]. Addressing breast cancer is a multi-front effort across the cancer control continuum, from prevention to treatment to survivorship. Great strides have been made in therapies and standards of care, leading to decreased mortality in developed countries. However, breast cancer incidence has remained essentially unchanged for the last three decades [2,3] indicating that a fresh approach to preventing breast cancer across the population is needed [4].

Given the lack of progress in breast cancer prevention, the California Breast Cancer Research Program (CBCRP) [5] applies current scientific knowledge about breast cancer to primary prevention at the population level. To turn the tide of breast cancer in the state, CBCRP supported the development of a Comprehensive Breast Cancer Primary Prevention Plan (BCPPP) for California [6]. The plan addresses all levels of the health impact pyramid, from education at the top to the bottom rungs of changing the context and socioeconomic factors, where the population impact is greatest [7]. The plan also considers risk factors at all stages of the lifespan. An overarching goal and specific intervention goals for 23 risk and protective factors are identified in the plan. These goals are defined with specific intervention strategies that could be used to reach these goals. Californians Linking Action with Science for Prevention of Breast Cancer (CLASP-BC) was conceptualized to translate these strategies into evidence-informed interventions (EIIs) that are disseminated and implemented across California.

CLASP-BC was informed by six years of funded dissemination and implementation (D&I) research and program evaluation conducted by the Canadian Partnership Against Cancer (CPAC) [8] through its Coalitions Linking Action and Science for Prevention (CLASP) [9,10]. CLASP was focused on creating supportive environments for cancer prevention through sustainable practice and policy change; and was implemented in two phases. Phase 1 engaged research, practice, and policy experts across Canada to identify key cancer prevention priorities [9], provided regional networking meetings in advance of calls for proposals to engage community-based organizations to identify barriers and enablers to participation in pan-Canadian coalitions, and provided preliminary feedback and guidance from the funder on an optional preliminary summary of a Phase 2 dissemination and implementation proposal [10].

CLASP-BC is designed to support the dissemination, implementation, and evaluation of evidence-informed intervention (EII) strategies from the BCPPP by leveraging existing community cancer and chronic disease prevention efforts and focusing on identified risk factors for breast cancer. CLASP-BC will also be implemented in two phases. As detailed in the future directions for research section of this paper, Phase 1 will focus on: (1) understanding the breast cancer concerns and prevention priorities of community leaders (from culturally diverse and medically underserved communities), researchers, practitioners, and policy experts across California; (2) engaging community and opinion leaders, research, practice, and policy specialists in regional California meetings to identify opportunities for working together in breast cancer prevention coalitions based on shared concerns and priorities; and (3) supporting (e.g., with technical assistance and training workshops) building, strengthening, and enhancing community-based participatory research (CBPR) and D&I research capacity, as well as research engagement within these coalitions. 

The conceptual framework underpinning the first phase of CLASP-BC incorporates the multi-disciplinary model to broaden participation in community problem solving through collaborative practice and research [11], concept mapping as a participatory public health research tool [12], community-based participatory research [13,14], and transforming power relations in coalition formation for community betterment and collaborative empowerment [15].

Following the implementation and evaluation of Phase 1, Phase 2 will focus on: (1) providing D&I research grant support for interested and eligible coalitions demonstrating in their funding applications collaborative, evidence-informed (both practice and science-based) breast cancer prevention approaches from the BCPPP across two or more California jurisdictions (e.g., cities, counties); (2) working through a cooperative agreement funding mechanism, successful applicants will share the knowledge gained on quarterly video conference calls and annual meetings to exchange ideas for how to meet the challenges and take advantage of the opportunities to sustain the breast cancer prevention approaches beyond the funding period: and (3) integrating the lessons learned from science with the lessons learned from practice and policy to reduce the risk of developing breast cancer.

CLASP-BC is designed to disseminate and implement high-impact, population-based prevention approaches by funding large scale EIIs, through multi-jurisdictional actions, with the intent to decrease the risk of breast cancer and other chronic diseases (sharing common risk factors), particularly among racial/ethnic minorities and medically underserved populations in California. Examples of medically underserved communities in California include black populations in counties such as Tuolumne and Mariposa counties where 56.3% and 49.1% of families respectively live below the poverty level; Alpine county where 62.5% of Hispanic families live below the poverty level; and Lake County and Glenn counties where 40.9% and 40.3% of Native American families respectively live below the poverty level [16]. Thus, given that one of the California Breast Cancer Primary Prevention Plan priorities is to promote post-natal breast feeding, it may be a community concern and priority that 42% of women living at or below the poverty level in California report having workplace breast feeding support compared to 83% of women living at 4 times the poverty level. Similarly, Latina (53%) and Black (59%) women report having workplace breast feeding support compared with 76% of white women [17].

In addition, engagement of indigenous communities may require working with tribal leaders and elders in ensuring acceptability of proposed partnerships. In California there are 109 federally recognized tribes [18]. Moreover, due to the large number of indigenous Latinos from Central and South America, California is also home to a large number of indigenous communities from Latin America (including Mixteco, Zapotec, among other). Language, culture and particular networks may need to be reached in culturally specific ways [19].

A comprehensive strategy to breast cancer prevention assumes that through multi-sector (government, community-based non-governmental organizations (NGOs), academia, and the private sector) and multi-jurisdictional approaches, working together will be more effective than when each organization, sector, or jurisdiction works on its own. By working together, partners share each other’s skills, knowledge, and resources, as well as the risks and rewards, to more quickly and effectively reach breast cancer and chronic disease prevention goals and objectives.

This paper describes in detail the plans for implementing CLASP-BC Phase 1. A separate paper in this special issue describes the Phase 2 elements of CLASP-BC predicated on the successful implementation and evaluation of Phase 1.

## 2. Background

### 2.1. Making the Case for Community-Based Partnerships

Much of the literature on coalition and partnership building [20,21,22] calls for the development of a common vision and mission, a clear understanding of the roles of different players, what resources they each bring to the table, and delineated strategies for coordination and sustaining the partnership. It appears that how much synergy is produced by the coalition members is what makes a difference often in how successful they become [23]. As resources become scarcer, a reliance on the building of community-based partnerships and well-established coalitions becomes ever more essential. Health equity coalitions provide us with important lessons in the building of CLASP-BC partnerships. Given the intersectionality factors of some of the populations with whom researchers need to collaborate, different types of organizations need to coalesce and be sustained over time in order to resolve common issues and community health problems.

One of the most important challenges faced today in social movements for change is how interactions and intersections of race, class, gender, sexuality, and power come together [24]. Community-based coalitions can be evaluated by the extent to which they are grounded in shared or overlapping interests, where groups identify a common ground and work together towards achieving these goals. However, for coalitions and partnerships today to be successful, there is also a need for capacity building within communities as well as scientific and public health institutions. Communities are challenged by many competing needs and with diverse populations where gender, race, sexuality and socioeconomic status may come together to create situations that place individuals at even greater disadvantage and challenge the ability of coalitions to meet partnership goals. Partnerships that are grounded in the community strengthen our communities by fostering the participation of key individuals and community leaders in the life of the community, engaging them in problem solving, and addressing issues, such as poverty, that impact both community residents and institutions [25]. They also result in community buy-in and often remarkable participation in the solutions that are crafted [26,27].

### 2.2. Community-Based Partnerships and Academia

Community-based partnerships and coalitions also provide benefits to scientific and public health institutions. They allow for the opportunity for researchers and academics to hear about specific barriers and knowledge gaps faced by various populations and regions. They also provide rich environments to move from theory and observation to deep contextual knowledge, which in turn improves theory. The demands of academia incentivize academic researchers to prioritize expediency, efficient scientific designs and publication over the qualities required for true collaboration with communities such as relationship- and trust-building, equitable service delivery, social change, advocacy, negotiation and recognition of the power imbalances between team members.

Negotiating this balance is not a skill taught in research training, and failure can lead to worsened academic-community relations. To successfully build relationships that foster collaboration and shared lived experiences requires time, listening to and understanding the needs and barriers of communities and how their environment may pose unique risks for increased breast cancer rates as well as opportunities to reduce risk. It also requires communities to have a clear understanding of academic needs, and to build skills on negotiating a common scientific agenda [28]. The information and experiences that can be learned through community partnerships can be invaluable towards identifying and implementing breast cancer prevention strategies.

Establishing strong community-based coalition partnerships that are responsive to community needs and priorities, and help to foster research by providing a good understanding of community needs [26], are necessary to achieve community health improvements. When done so communities and academia are able to work together, achieving societal agreement on their priorities and objectives [29]. For example, a 2011 study examined changes in cancer disparities in a community partnership network, including 20 local community partner organizations in Southwest Florida. A social network analysis identified a trend toward increased network decentralization and overall increases in linkages, which supported the sustainability of the network and stability over a three-year period [30].

### 2.3. Creating Meaningful and Culturally Appropriate Partnerships

Mechanisms and principles for building multisector partnerships for population health and health equity that can be sustainable have been described [31]. While only 10% of public health measures tracked are met, this is due in part to the lack of responsibility and ownership for the outcomes, lack of cooperation and collaboration, and a limited understanding of what really might work or not in a particular community. In order for coalitions to have an impact, there are several elements that must be considered. These include: (1) the proper analysis of information, (2) recognition of community knowledge as critical information, (3) the establishment of a common vision and mission, (4) shared learning, (5) the ability and dedicated efforts towards using strategic and action plans, and (6) the development of effective leadership among them. Mechanisms by which one can document progress with a feedback loop back into the process, and looking towards identified outcomes, can strengthen partnerships that will achieve goals in population health and health equity.

One key finding that permeates many of these studies is the factor of “synergy.” Partnership synergy, combining the individual perspectives, resources, and skills of coalition partners to create something new and valuable, has been examined and a practical framework developed for studying and strengthening the collaborative advantage [32]. Mechanisms were conceptualized that account for the collaborative advantage in order to strengthen the capacity of partnerships in the future to realize their full potential. Martinez-Bianchi et al. [33] addressed the capacity of family medicine to improve health equity through collaboration, accountability and coalition building. To achieve health equity an evaluation of social, economic, environmental and other factors that impede optimal health for families are needed. Thus, coalitions partner not only with communities, but also with families. A Health Equity Toolkit [33] was developed, designed to improve care systems, reduce disparities and improve patient outcomes. Engaging with government and community-based non-governmental agencies, academic centers and the private sector are needed to create partnerships to systematically tackle health inequities.

### 2.4. Community Engagement

To engage diverse constituencies at meetings, they must be accessible to all participants and be perceived as worth the investment in time. Some best practices include offering translated materials and simultaneous interpretation upon request, facilitating a safe space for all participants. Facilitating a safe space means cultivating and upholding community agreements and inviting all participants and facilitators to share their pronouns. Events and venues must be accessible and centrally located for various members with sufficient parking or public transit access. Consideration of the times of the day when members meet, may be necessary, as well as offering food, parking, and child-care on the premises.

During COVID-19, connection is necessarily virtual and remote, creating barriers for many underserved and vulnerable communities that lack access to reliable Wi-Fi and the necessary technology. These barriers may further limit access to events, complicate outreach efforts and pose challenges to community partners joining virtual meetings. To overcome these barriers, event organizers should send out meeting information ahead of time, by mail if necessary, and by email, including simple instructions for joining on-line platforms and telephone call-in options. Creating accessible meetings allows for more intersectional attendance and creates a space for individuals to share their lived experiences. Community members’ stories highlight the complex intersections that can contribute to breast cancer risk factors and are an essential tool for tailoring breast cancer prevention efforts to the needs of different communities. Additionally, community involvement and coalition-building create another avenue to share information about breast cancer prevention and empower community members to share information within their communities that are culturally and linguistically appropriate.

CLASP-BC is expected to have wide reach and involvement across the entire public health and medical spectrum of those engaged with breast cancer. It is expected that participants involved in outreach, engagement, evaluation and/or dissemination of the activities of this program may include, but are not necessarily limited to: practitioners, public health specialists, community health workers, citizen scientists, patient advocates, patients and their families, social service agency leaders, policy makers, opinion leaders, business leaders, civic leaders, researchers, cancer centers, cancer registries (e.g., NCI’s Surveillance, Epidemiology, & End Results (SEER) program, local hospitals, local clinics, local public health agencies, governmental and non-governmental organizations, community based organizations, academic units, regional municipalities, jurisdictions and district offices, coalitions, faith-based organizations, and interested citizens who can form part of data-gathering, strategies and solutions proposed.

### 2.5. Lessons Learned from the Canadian CLASP Experience

There are both similarities and significant differences between CPAC’s CLASP initiative and the proposed CLASP-BC initiative. In terms of the population focus, while Canada (37.6 million in 2019) and California (39.8 million in 2019) have similar sized populations, they have radically different population densities (Canada 3 per square mile, California 251 per square mile). Both Canada and California have high percentages of their populations living in urban areas (80% in Canada, 95% in California). In California, 39% of state residents are Latino, 37% are white, 15% are Asian American, 6% are African American, 3% are multiracial, and fewer than 1% are American Indian or Pacific Islander, according to the 2018 American Community Survey. Latinos surpassed whites as the state’s single largest ethnic group in 2014 [34]. In Canada, ethnic categories are not comparable to USA racial/ethnic categories [35].

CPAC’s pan-Canadian CLASP framework focused on engaging communities from 10 Canadian provinces and 3 territories. The proposed CLASP-BC initiative would be focused on engaging communities across the state (e.g., 58 California counties, 482 municipalities) prioritizing those communities with the highest percentage of racially and ethnically diverse and medically underserved populations that bear the greatest burden of breast cancer diagnoses.

Throughout the CLASP experience in Canada, two key challenges were encountered consistently with respect to community engagement:

#### 2.5.1. Community-Based Partner Capacity

In providing multidisciplinary knowledge to action research opportunities, a key learning from the Canadian CLASP initiative was the importance of considering equity in capacity across the partner disciplines [36]. Research and policy partners were well positioned to respond to the request for Phase 2 proposals by leveraging their experience in proposal writing, contract management, and reporting. Conversely, community-based practice organizations often lacked the experience and/or capacity to pursue complex funding opportunities but brought unique and important community perspectives and relationships to coalitions that otherwise would have lacked these elements. This was particularly true of First Nations, Inuit, and Métis community partners.

#### 2.5.2. Indigenous Engagement

Many coalitions included First Nations partners—either organizations or communities—and a key challenge was fostering respectful and meaningful engagement and aligned partnerships [21]. Indigenous communities frequently had specific governance structures and protocols for approvals and communication that differed from many academic and government processes. If the other partners were not experienced in working with Indigenous communities, this misalignment was often a source of confusion and misunderstanding. In addition, building cultural sensitivity in partners who were not well-versed in working with Indigenous communities was important to ensure respectful engagement and building trust in the context of hundreds of years of past trauma.

## 3. Future Directions for Research

The overall purpose of CLASP-BC is to link the lessons learned from science (Knowledge to Action) with the lessons learned from practice and policy (Action to Knowledge). In doing so, it will broaden the reach and deepen the impact of EIIs on breast cancer and chronic disease prevention initiatives across California, focused on common risk factors, and will expand their impact. The foci of Phase 1 are building coalitions, coalition capacity building through community engagement, and cooperative and collaborative CBPR and dissemination and implementation research training. The specific aims of phase 1 are as follows:
In light of the COVID-19 pandemic, in the California context, conduct pre-planning prior to capacity building to determine when the timing is appropriate to engage community partners in coalition capacity building focused on breast cancer prevention and how best to engage community participants if the opportunities for face-to-face meetings remain limited.Engage community leaders, including elected officials, from culturally diverse and medically underserved California communities to understand their breast cancer prevention priorities in relation to other community health priorities (including COVID-19).Engage research, practice, advocacy and policy specialists to understand their breast cancer prevention priorities in relation to the dissemination and implementation of evidence informed interventions.Foster equitable coalition partnerships to conduct successful CBPR/D&I research by providing training and technical assistance to:
(a)develop and sustain community, research, practice, and policy partnerships and conduct research using CBPR methods.(b)create a shared vision for partnerships.(c)develop model team agreements and tools for conducting partnership self-assessments.(d)identify benefits and challenges of participating in CBPR for their own partnership and identify potential solutions to challenges.Assist teams in creating a pathway from vision to research project consideration by providing training and technical assistance to:
(a)identify the various scientific methods that may be used in innovative intervention/adaptation research leading to D&I research.(b)identify community concerns and potential solutions regarding which research methods best fit their contexts.(c)identify and select best practices that will be utilized in creating a strong (qualitative and quantitative) and sustainable measurable methods plan, using theoretical frameworks.

For the purposes of CLASP-BC, the definitions of community representatives and patient advocates, as well as research, practice, and policy experts, who will be invited to participate in the meetings described below, are as follows:

Community representatives and patient advocates—these are individuals who live and work in the engaged communities and/or are leaders in community-based organizations providing vital social, economic and health service support in the engaged communities. As such, these coalition partners are vital in contributing their knowledge and expertise as community leaders. In Canada, they often referred to as community champions. US examples are “promotores de salud” (community health workers that have won the trust of their communities), serving as citizen scientists who are engaged in supporting research at different levels in the community. These individuals can be trained in research and can be engaged from conceptualization through implementation, evaluation and dissemination of findings for research and practice projects.

Research experts—individuals with an advanced degree (e.g., Masters or Doctorate) who have actively participated in and contributed to the research enterprise as evidenced by peer-reviewed research grants and/or publications. Researchers who have such a demonstrated research background may or may not be affiliated with an academic institution (e.g., academic cancer centers) but could also serve in an NGO, government, or other organizations with research as part of their mission.

Practice experts—individuals who manage and/or provide programming and/or services that influence directly or indirectly (e.g., built environment) population health. Practitioners in the funding agreement applications could represent NGOs, government, or other organizations with demonstrated knowledge and skill in the topic under consideration for the funding application.

Advocacy and policy experts—individuals who work on making or influencing policy decisions in or outside of government (e.g., an NGO) that influence directly or indirectly population health. Policy can include legislative or executive decisions that work through taxation, regulation, and related policy instruments that impact populations. This may also include advocacy groups that advocate for informed policy decision making and help to educate and inform policy-makers, disseminating scientific findings.

In Phase 1, interested parties will be invited to participate in a state-wide concept mapping [12] activity focused on identifying community, research, practice, and policy priorities for breast cancer prevention that align with other broader public health priorities (e.g., increasing physical activity, reducing exposure to environmental pollutants). Concept mapping is a mixed methods approach that engages diverse stakeholders through a multistep process to generate ideas, organize them into distinct categories, and rate them according to a set of criteria (e.g., feasibility, importance). All community, research, practice, and policy participants in the concept mapping exercise will be invited to indicate whether or not they are interested in participating in regional one-day concept mapping results review and coalition engagement meetings. In addition, potential co-funding organizations for phase 2 D&I research grants will be invited to attend these one-day meetings. The results of the concept mapping analyses will be used to organize multiple one-day regional meetings (in person and/or online) across California inviting participant constituencies that share similar breast cancer prevention priorities to attend.

Subsequently, two-day CBPR and D&I training workshops will be organized for existing or emerging coalitions interested in applying to a future Phase 2 D&I research funding initiative. These meetings will provide CBPR and D&I research tools and guidance as well as coalition exercises to help potential applicants develop a shared common understanding of the opportunities and challenges of working together collaboratively on EII for breast cancer prevention.

### CBCRP Support for CBPR and D&I Research Training and Provision of Technical Assistance

For the past decade, CBCRP has conducted NIH-supported proposal development workshop programs to help new and existing community-academic teams build partnerships and develop fundable research projects focused on breast cancer environmental causes and disparities. Based on this experience, the following draft (see Box 1) of the proposed two-day CLASP-BC CBPR/D&I training workshop is described below. This workshop draft will be further refined based on input gathered from the concept mapping data prior to, and the evaluation data collected during, the aforementioned one-day regional engagement meetings.

Box 1Draft CBCRP/D&I Training Workshop.
**Topic and Learning Objectives**
**The California Plan to Prevent Breast Cancer**
[6,7,37]

*Learning Objectives:*

Understand Concepts of Population Primary PreventionUnderstand the complexity of breast cancer causation and methods to reduce incidence
**Introduction to Implementation Science**
[38,39,40]

*Learning Objectives*

Understand broad objectives of the fieldDefine and understand key terminologyUnderstand types of research questionsIdentify what is (and is not) implementation science
**Finding Your Research Question and Framing Your Questions**
[41,42]

*Learning Objectives*

Learn the elements of effective research questionsUnderstand the key components of specific aims in implementation science

**Small Group Activity**
Defining the problem, crafting the research question and identifying aims
**Engaging Your Community and Forming Your Coalition**
[43,44]

*Learning Objectives*

Learn the concentric circles of community engagementLearn how to use best practices and tools to develop coalitions

**Small Group Activity**
**Theories, Frameworks, and Models**
[45,46]

*Learning Objectives*

Understand the importance and role of theories, frameworks, and modelsUnderstand how these theories, frameworks, and models inform study components
**CBPR in Practice**
[47]

*Learning Objectives*

understand the benefits and challenges of CPPR for both the community and academic partnersbe able to identify benefits and challenges of participating in CPPR for their own partnershiplearn methods to mitigate barriers
**Study Designs in Dissemination and Implementation Research**
[48,49,50]

*Learning Objectives*

Learn about study design options for implementation scienceUnderstand basic measures, outcomes, and analyses in implementation science
**Measurement, Epidemiology, and Evaluation**
[51,52,53,54,55]

*Learning Objectives*

Understand the importance of measurement and evaluation issuesConsider measurement issues in designing studiesLearn about resources for selecting implementation science measures

**Small Group Activity**
Choosing a study design, defining outcomes and measures

**Tools and Technical Assistance**

**Conclusion**


All potential Phase 2 D&I research coalition applicants (including the lead research, practice, policy and community experts of each coalition team) will be invited to attend the CBCRP-sponsored two-day CBPR & D&I research training workshops described above.

In addition to the peer-reviewed publication of Phase 1 meetings data, media and social media presentations of the concept mapping findings by region will be developed and disseminated to increase interest and possible in engagement in Phase 2 CBPR/D&I research coalitions. In addition, CBCRP will share the findings from the concept mapping and regional meetings with potential Phase 2 co-funding agencies which are unable to attend all or any of the regional engagement meetings as observers.

## 4. Application to Breast Cancer Prevention

The California Breast Cancer Primary Prevention Plan (BCPPP) [6], was based on a strong foundation of science and input from many stakeholders, including academics, government regulators, non-profit organizations and impacted communities, and developed a policy agenda and action plan, to reduce the incidence of breast cancer in the state. Over the two-year project, BCPPP held a series of webinar-based study groups to:
explore the strength of the science behind known and suspected risk factors for breast cancer;explore potential interventions to address these risk factors;identify strengths, weaknesses and gaps in scientific research; andwork with the broad array of stakeholders to disseminate and implement the plan.

The process was guided by a multi-stakeholder advisory committee that included some of California’s leading breast cancer, public health, social and environmental justice and disease prevention experts. The project culminated in the creation of the Comprehensive Breast Cancer Primary Prevention Plan for California, which will serve as a road map for legislators, local and state regulators, and community.

The plan outlines a series of 16 overarching goals along with specific interventions that would support the accomplishment of these goals. The following are selected examples of intervention strategies from the Plan:
Update state, city and county zoning and permitting laws, as well as city and county General Plans, to prevent polluting industries from being located near schools or concentrated in communities of color or low-income communities.Develop safe walk-, bike- and public transit-friendly cities to enhance physical activity opportunities and reduce pollution, both of which impact breast cancer risk and health in general.Create breastfeeding-supportive workplaces for all workers, regardless of employment classification or status.Establish standards of best practices for all occupations where workers might be exposed to ionizing radiation.Develop workplace policies, with worker involvement, to reduce, eliminate or mitigate unnecessary exposures to light at night.In accordance with California’s recognition of the human right to water, expand the state’s capacity to ensure safe (free from chemicals linked to breast cancer), adequate and affordable water for all California residents, regardless of whether they live in cities, towns or unincorporated areas.

Evidence-informed interventions (EIIs) to prevent breast cancer linked to these strategies include but are not limited to: limiting alcohol intake; promoting physical activity, healthy eating for healthy weight [56]; promoting postnatal breastfeeding [57]; mitigating the impact of night shift work [58]; minimizing exposure to all forms of ionizing radiation, particularly among girls and young women; ensuring that consumer products are free from chemicals linked to breast cancer [59]; eliminating unnecessary exposure to circadian rhythm-disrupting light at night [60]; and reducing the use of exogenous hormones [61]. Each of these EIIs should be adapted, disseminated, implemented and evaluated in a manner that addresses historical discrimination and oppression based on race, ethnicity, gender identity and orientation, sexual orientation, immigration status, disability, or other factors that may affect breast cancer risk.

Helping to operationalize the intervention strategy priorities identified in the BCPPP, and providing resources for community-level interventions, guidelines are published by many governmental and non-governmental agencies. These resources use an evidence-informed process and consensus, with varying degrees of rigor. For example, the Guide to Community Preventive Services (the Community Guide) is a systematic review that summarizes what is known about the effectiveness and cost-effectiveness of population-based interventions designed to promote health, prevent disease, injury, disability and premature death [62]. Related efforts such as the National Cancer Institute’s Research-tested Interventions Programs (RTIPs) provide cancer control practitioners access to over 200 programs that had been evaluated, have shown positive outcomes, and are published in peer-reviewed journals [63]. A unique focus of RTIPs is that programs are rated across the Reach, Effectiveness, Adoption, Implementation, & Maintenance (RE-AIM) framework [64] to assess the potential for implementation and long-term impact. Other guidelines such as What Works for Health rely on a wider range of standards of evidence, including expert opinion [65].

While policy dissemination research is relatively under developed in the field of health, policy dissemination research in other areas is not a new field and is more developed in countries outside the United States [66]. The lessons learned from this research, in specific California jurisdictions, as appropriate, will be translated across the state and may be replicable in other jurisdictions outside the state of California.

## 5. Conclusions

This paper describes a planned research funding initiative by the California Breast Cancer Prevention Program entitled Californians Linking Action with Science for Prevention of Breast Cancer (CLASP-BC). CLASP-BC is informed by CBPR and D&I research and is designed to translate the California Breast Cancer Prevention Plan overarching goal and specific intervention goals for 23 breast cancer risk and protective factors strategies into evidence-informed interventions (EIIs) that are disseminated and implemented across California. CLASP-BC is designed to disseminate and implement high-impact, population-based prevention approaches by funding large scale EIIs, through multi-jurisdictional actions, with the intent to decrease the risk of breast cancer and other chronic diseases (sharing common risk factors), particularly among racial/ethnic minorities and medically underserved populations in California.

CLASP-BC was informed, in part, by the Coalitions Linking Action and Science for Prevention (CLASP) D&I research and program evaluation initiative funded by the Canadian Partnership Against Cancer (CPAC). Like the Canadian CLASP, CLASP-BC will be rolled out in two phases. Phase 1 will focus on: (1) understanding the breast cancer concerns and prevention priorities of community leaders (from culturally diverse and medically underserved communities), researchers, practitioners, and policy experts across California; (2) engaging community and opinion leaders, research, practice, and policy specialists in regional California meetings (virtual and/or in-person) to identify opportunities for working together in breast cancer prevention coalitions based on shared concerns and priorities; and (3) supporting (e.g., with technical assistance and training workshops) building, strengthening, and enhancing CBPR and D&I research capacity, as well as research engagement within these coalitions.

Phase 2 will focus on: (1) providing D&I research grant support for interested and eligible coalitions demonstrating in their funding applications collaborative, evidence-informed (both practice and science-based) breast cancer prevention approaches from the BCPPP across two or more California jurisdictions (e.g., cities, counties); (2) working through a cooperative agreement funding mechanism, successful applicants will share the knowledge gained on quarterly video conference calls and annual meetings to exchange ideas for how to meet the challenges and take advantage of the opportunities to sustain the breast cancer prevention approaches beyond the funding period: and (3) integrating the lessons learned from science with the lessons learned from practice and policy to reduce the risk of developing breast cancer.

This paper describes in detail the plans for implementing CLASP-BC Phase 1. The Phase 2 elements of CLASP-BC, predicated on the successful implementation and evaluation of Phase 1, are described elsewhere in this special issue.

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
