# Peer review of "Doing What We Know, Knowing What to Do: Californians Linking Action with Science for Prevention of Breast Cancer (CLASP-BC)"

_ijerph, 2020, doi:10.3390/ijerph17145050_

Round 1

Reviewer 1 Report

[Introduction]

Page 3, line 104-106. The CLASP-BC aims to target on racial/ethnic minorities and medically underserved populations in California. It is unclear which racial/ethnic minorities are this study targeting on.

[Background]

  1. Please include theories or conceptual framework that is underpinning the study.
  2. This section has several subheadings including “Community Engagement.” I recommend dividing the paragraphs in page 3-4 using subheadings, as well.
  3. Page 5, subheading ”Lessons learned from the Canadian CLASP experience” : I recommend adding information on population characteristics of California. How many and which racial/ethnic minorities are living in California? Which minorities would be included in this study?

[Future Directions for Research]

  1. Page 7, lines 284-308: Please consider moving these sentences (definitions of ‘terms’) under “Background” section.

[Summary]

  1. Page 11, lines 427-435: “Introduction” has the same sentences. Please paraphrase.

Author Response

Thank you for your comments and helpful suggestions. Please see our responses in italics below your feedback

Page 3, line 104-106. The CLASP-BC aims to target on racial/ethnic minorities and medically under-served populations in California. It is unclear which racial/ethnic minorities are this study targeting on.

Please see lines 116 -130 in the revised manuscript.

[Background]

  1. Please include theories or conceptual framework that is underpinning the study. Please see lines 96-100 in the revised manuscript.

  1. This section has several subheadings including “Community Engagement.” I recommend dividing the paragraphs in page 3-4 using subheadings, as well. Please see lines 141,166, and 191 in the revised manuscript.

  1. Page 5, subheading ”Lessons learned from the Canadian CLASP experience” : I recommend adding information on population characteristics of California. How many and which racial/ethnic minorities are living in California? Which minorities would be included in this study? Please see lines 252 – 256 in the revised manuscript.

Reviewer 2 Report

The paper describes the importance of an interdisciplinary approach to health and focuses on communication between groups and policy change. I support a proposal as described in this paper but think it will have its challenges as the bigger the number of people in the project, the harder it is for the different groups to communicate and coordinate their efforts. This type of recommendation is beyond the scope of this review but I have recommended some minor changes.

Abstract

The paper is written as a project proposal and this initial phase seems to focus on the policy, communication and formation of coalitions however, the abstract says "apply current scientific knowledge about breast cancer to primary prevention" While this is a goal of the initiative, it is not evident at this phase. Line 23 in the abstract states "specific intervention goals for 23 breast cancer risk and protective factors" but it is not clear where these are located in this proposal. The abstract also makes note of "other chronic diseases (line 38) but this concept seems to be lost within the context of this paper.

I would recommend that the authors review the abstract and ensure it only includes objectives that are included within this publication unless it is clearly stated otherwise.

Introduction

Line 63 - The statement "Change education and context and socioeconomic factors” are very broad, general terms which seem to be outside the objectives of this publication.

Line 63 - what is meant by "context" please define briefly

Line 70 - Is "informed" the correct term to use here?

Lines 83-90 it outlines the 3 main objectives of phase 1 which are this paper. It may be beneficial to more clearly indicate where these objectives are addressed within the paper.

Background

Line 118 - after the word table, should a reference be included here as it is talking about the literature?

Number the points starting at line 167.

Line 173-174 -  please define “partnership synergy" for the less knowledgeable readers outside the field.

 Line 181-182 If a reference required to the statement that a tool was developed, if it was not developed by the authors?

Future directions

Line 268 – this is a very general statement. I do not know if it could be made more applicable to the phase 1 objectives?

Line 282-308 – a number of terms are defined which is good and useful but it is not clear why these terms needed defining at this time. It needs a statement of context.

Line 334 – the workshop is good and so are the titles but some of the objectives are so general that it is difficult to appreciate the content Ex. “Understand broad objectives of the field” (define the field) and “learn methods to mitigate barriers” (define the barriers). Understand the importance of measurement and evaluation issues (issues??)

 Application to Breast cancer prevention

Line 357-358 – Should a reference be included for the plan that is discussed?

Summary

No recommendations for change.

Author Response

Thank you for your detailed comments and suggestions.  Please see our replies in italics listed below your feedback.

Abstract

The paper is written as a project proposal and this initial phase seems to focus on the policy, communication and formation of coalitions however, the abstract says "apply current scientific knowledge about breast cancer to primary prevention" While this is a goal of the initiative, it is not evident at this phase. Line 23 in the abstract states "specific intervention goals for 23 breast cancer risk and protective factors" but it is not clear where these are located in this proposal. The abstract also makes note of "other chronic diseases (line 38) but this concept seems to be lost within the context of this paper.

I would recommend that the authors review the abstract and ensure it only includes objectives that are included within this publication unless it is clearly stated otherwise.

Please see revisions to the abstract as per your recommendation.

Introduction

Line 63 - The statement "Change education and context and socioeconomic factors” are very broad, general terms which seem to be outside the objectives of this publication.

To turn the tide of breast cancer in the state, CBCRP supported the development of a Comprehensive Breast Cancer Primary Prevention Plan (BCPPP) for California.   The plan addresses all levels of the health impact pyramid, from education at the top to the bottom rungs of changing the context and socioeconomic factors, where the population impact is greatest.”  

These terms are a specific reference:

White MC, Kavanaugh-Lynch MMHE, Davis-Patterson S, Buermeyer N. An Expanded Agenda for the Primary Prevention of Breast Cancer: Charting a Course for the Future. Int J Environ Res Public Health. 2020;17(3):E714. Published 2020 Jan 22. doi:10.3390/ijerph17030714

to how the BCPPP was based in part on the health impact pyramid:

Frieden TR. A Framework for  Health Action: The Health Impact Pyramid. American Journal of Public Health. 2010;100(4):590-595. doi:10.2105/ajph.2009.185652.

As such, it does not specifically refer to CLASP-BC.

Line 63 - what is meant by "context" please define briefly

In the referenced Frieden publication, context is introduced as follows:

“Changing the Context to Encourage Healthy Decisions The second tier of the pyramid represents interventions that change the environmental context to make healthy options the default choice, regardless of education, income, service provision, or other societal factors. The defining characteristic of this tier of intervention is that individuals would have to expend significant effort not to benefit from them. For example, fluoridated water—which is difficult to avoid when it is the public supply—not only improves individual health by reducing tooth decay,22 but also provides economic benefits by reducing health spending and productivity losses. In countries without either adequate natural or added fluoridation, health authorities are limited to counseling interventions, such as encouraging toothbrushing.”

Line 70 - Is "informed" the correct term to use here?

Evidence-based practices and programs may be described as "supported" or "well-supported", depending on the strength of the research design. Evidence-informed practices use the best available research and practice knowledge to guide program design and implementation. Given that intervention strategies described in the BCPPP will perforce need to be adapted to local community contexts, we believe EII is the correct term.

Lines 83-90 it outlines the 3 main objectives of phase 1 which are this paper. It may be beneficial to more clearly indicate where these objectives are addressed within the paper.

Please see lines 87-95:  As detailed in the future directions for research section of this paper, of the revised manuscript,  Phase 1 will focus on:  1) Understanding the breast cancer concerns and prevention priorities of community leaders (from culturally diverse and medically under-served communities), researchers, practitioners, and policy experts across California; 2) Engaging community and opinion leaders, research, practice, and policy specialists in regional California meetings to identify opportunities for working together in breast cancer prevention coalitions based on shared concerns and priorities; and 3) Supporting  (e.g., with technical assistance and training workshops) building, strengthening, and enhancing community-based participatory research (CBPR) and D&I research capacity, as well as research engagement within these coalitions.

Background

Line 118 - after the word table, should a reference be included here as it is talking about the literature?

Please see lines 141-144 in the revised manuscript.

Number the points starting at line 167. Please see lines 197-199.

Line 173-174 -  please define “partnership synergy" for the less knowledgeable readers outside the field.

Please see lines 204-205 in the revised manuscript.

Line 181-182 If a reference required to the statement that a tool was developed, if it was not developed by the authors?  

Please see line 212 in the revised manuscript.

Future directions

Line 268 – this is a very general statement. I do not know if it could be made more applicable to the phase 1 objectives?

Here we are referring to creating a shared vision for partnership by meeting participants.  We have added the word shared to the description. Please see line 306 in the revised manuscript.

Line 282-308 – a number of terms are defined which is good and useful but it is not clear why these terms needed defining at this time. It needs a statement of context.

We have added the following phrase to the sentence prior to the term definitions:  “who will be invited to participate in the meetings described below” Please see lines 321-322 in the revised manuscript.

 Line 334 – the workshop is good and so are the titles but some of the objectives are so general that it is difficult to appreciate the content Ex. “Understand broad objectives of the field” (define the field) and “learn methods to mitigate barriers” (define the barriers). Understand the importance of measurement and evaluation issues (issues??)

As noted prior to the table outlining the workshop, the workshop draft will be further refined based on input gathered from the concept mapping data prior to, and the evaluation data collected during, the aforementioned one-day regional engagement meetings. As such, we don’t think that further specification of the content is advised until we have completed the concept mapping and regional engagement meetings.

Application to Breast cancer prevention

Line 357-358 – Should a reference be included for the plan that is discussed? Please see line 385 in the revised manuscript.

Summary

No recommendations for change.